# Towards Zero-shot Learning
# for End-to-end Cross-modal Translation Models

**Jichen Yang**[1*]**, Kai Fan**[2* †]**, Minpeng Liao**[2*]**, Boxing Chen**[2]**, Zhongqiang Huang**[2]
[1]The Hong Kong Polytechnic University
[2]Alibaba DAMO Academy
jichen.yang@polyu.edu.hk
{k.fan,minpeng.lmp,boxing.cbx,z.huang}@alibaba-inc.com

## Abstract

One of the main problems in speech translation is the mismatches between different modalities. The second problem, scarcity of parallel data covering multiple modalities, means that the end-to-end multi-modal models tend to perform worse than cascade models, although there are exceptions under favorable conditions. To address these problems, we propose an end-to-end zero-shot speech translation model, connecting two pre-trained uni-modality modules via word rotator's distance. The model retains the ability of zero-shot, which is like cascade models, and also can be trained in an end-to-end style to avoid error propagation. Our comprehensive experiments on the MuST-C benchmarks show that our end-to-end zero-shot approach performs better than or as well as those of the CTC-based cascade models and that our end-to-end model with supervised training also matches the latest baselines.

## 1 Introduction

Speech translation (ST) requires knowledge transfer among different modalities, whereas models more often than not perform worse on cross-modal tasks. The ST model in real-world applications is usually a cascade approach that first uses an automatic speech recognition (ASR) system to transcribe the speech into text and then uses a text machine translation (MT) model. Recent end-to-end (e2e) ST models remove the need for an explicit ASR, with several practical advantages over the cascade models such as reduced latency, reduced error propagation, and shorter pipeline.

However, e2e ST models are less competitive than cascade models in practice (Zhang et al., 2019; Sperber and Paulik, 2020; Dinh, 2021) because end-to-end data are an order of magnitude less than those for ASR or MT, especially for low-resource language pairs. Solutions have been proposed to combat this data problem. In (Liu et al., 2020; Xu et al., 2021), an adapter with additional parameters is used during fine-tuning to combine the two pre-trained models of different modalities. The new module, however, only learns from ST data, which is of a greatly reduced quantity. The alignment in building cross-modal representations is also a popular topic. Zhang et al. (2023) simply concatenates the representations of different modalities and lets the self-attention learn the cross-modal features. Some solutions deal with this problem through mapping features into fixed-size representations (Reimers and Gurevych, 2019; Feng et al., 2020; Han et al., 2021; Duquenne et al., 2022). The squared error is generally used as the optimization objective (Pham et al., 2019; Dinh et al., 2022). They may suffer from information loss when representations are compressed or constrained prior.

In order to overcome both the data and the length problems, we propose a pre-trainable adapter that connects two pre-trained modules. Specifically, we adopt a popular cross-modal ST architecture that can be generalized to many existing works. For the alignment adapter, we employ as loss the Word Rotator's Distance (WRD) minimization (Yokoi et al., 2020; Monge, 1781; Kantorovich, 1960; Peyré et al., 2019), allowing the adapter to promote the cross-modal representations that match in the space of the semantic encoder. Unlike previous works, this strategy allows us to pre-train the adapter. Meanwhile, instead of mapping to a fixed length, the CTC enables adjustment of the length of the source modality representation dynamically. This step can guarantee the cross-modal representations become features with a similar but not exactly the same length, and then our proposed WRD objective with optimal transport (OT) solver can align them properly.

Besides speech translation, our approach can be naturally adapted into image translation. Unlike the

---

*equal contribution. Work was done during Jichen Yang's research internship at DAMO Academy, Alibaba Group.
† Corresponding author.

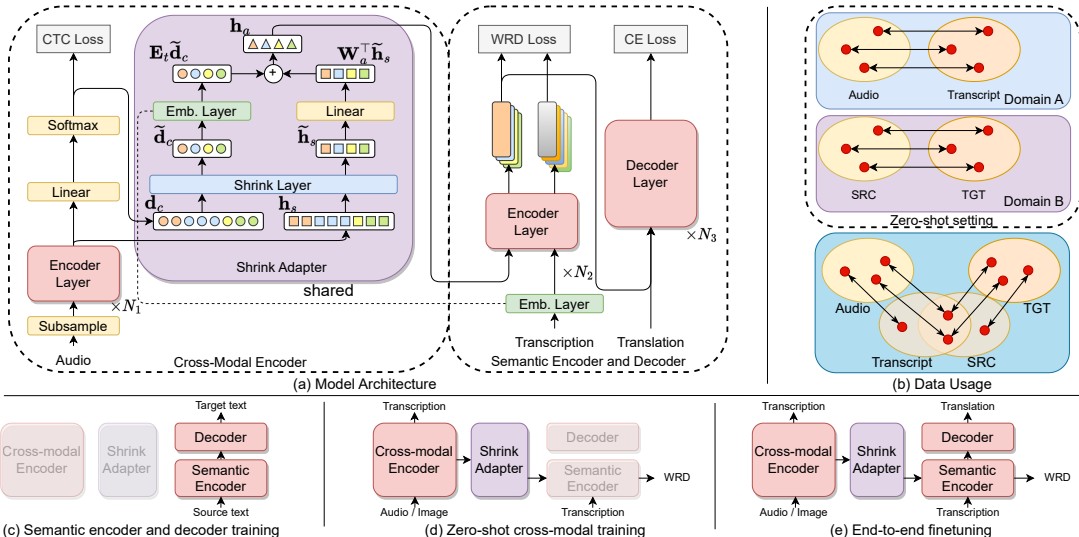

Figure 1: Overview of our proposed framework. **(a)** The overall model architecture. **(b)** Data usage in zero-shot setting and the possible supervised setting. **(c)** NMT pre-training. **(d)** Only the cross-modal encoder is trained with ASR data, where the semantic encoder is freezing. **(e)** Fine-tuning if end-to-end data is available.

image assistant translation (Su et al., 2019; Yawei and Fan, 2021), we attempt to translate the text within the image. Our goal is also to align the cross-modal representation between images and texts (Li et al., 2022). The related discussion and experiments can refer to the Appendix A.4.

The contributions of this paper are as follows:
**(1)** We adopt the WRD loss together with the shrink mechanism to measure two feature sequences in different lengths, enabling the adapter pre-training.
**(2)** The pre-trained adapter allows an end-to-end zero-shot ST ability like cascade models.
**(3)** Experiments on the MuST-C demonstrate that our end-to-end zero-shot model can match or be slightly better than the CTC-based cascade model (without intermediate post-processing). The results of our end-to-end training can also match the recent supervised ST baselines.

## 2   Main Method

We adopt a popular framework in Figure 1(a), including a cross-modal encoder with a shrink adapter and a semantic encoder/decoder pack.

### 2.1   Semantic encoder-decoder training

A machine translation model is first pre-trained as illustrated in Figure 1(c). Given a machine translation corpus $\mathcal{D}_{mt} = \{(\mathbf{x}_t, \mathbf{y}_t)\}$, our aim is to obtain a semantic encoder $\mathrm{Enc}_t(\mathbf{E}_t\mathbf{x}_t) = \mathbf{h}_t$ and a semantic decoder $\mathrm{Dec}_t(\mathbf{h}_t) = P(\mathbf{y}_t|\mathbf{h}_t)$, where $\mathbf{E}_t$ is the source embedding matrix. The objective of the task is defined as the cross entropy loss $\mathcal{L}_{mt}$.

### 2.2   Zero-shot Translation Training

In this phase, we train a zero-shot translation model by training the cross-modal encoder alone as shown in Figure 1(d). As the tradition, we apply the recognition task with ASR data $\mathcal{D}_{cm} = \{(\mathbf{z}_s, \mathbf{x}_s)\}$, and adopt a classical ASR architecture with CTC classifier, and optimize the CTC loss $\mathcal{L}_{ctc}$.

Besides the regular recognition task, we use Word Rotator's Distance (WRD) (Yokoi et al., 2020) to supervise the encoder to generate encoding results with less discrepancy across different modalities. We expect to align different modalities in the space of the semantic encoder, allowing the seamless transition between the cross-modal encoder and the semantic decoder. To be precise, suppose the acoustic encoder output as $\mathbf{h}_s$ and the CTC distribution as $\mathbf{d}_c = \mathrm{softmax}(\mathbf{W}_c\mathbf{h}_s)$, a lightweight adapter shrinks and integrates them. The shrink mechanism (Yi et al., 2019; Tian et al., 2020; Gaido et al., 2021) is widely employed to remove the representations of blank and repeated tokens. Thus, we consider using the CTC path via efficient $\arg\max$ as the guidance to remove the blank tokens and average the representations of consecutively duplicated tokens, as shown in Figure 1(a).

By denoting the shrunk hidden state and CTC distribution as $\widetilde{\mathbf{h}}_s$ and $\widetilde{\mathbf{d}}_c$, the adapter output is,

$$\mathbf{h}_a = \mathbf{E}_t\widetilde{\mathbf{d}}_c + \mathbf{W}_a^\top\widetilde{\mathbf{h}}_s, \qquad (1)$$

where $\mathbf{W}_a$ is the trainable parameters in the adapter. More details can refer to the implementation. Since

| Training Data | En-De v2 | | En-De | | En-Fr | | En-Es | |
|---|---|---|---|---|---|---|---|---|
| | common | he | common | he | common | he | common | he |
| WMT | 28.59 | 27.45 | 28.62 | 27.44 | 40.79 | 37.54 | 32.38 | 38.14 |
| WMT + MuST-C | 33.13 | 31.99 | 32.99 | 31.90 | 44.14 | 39.97 | 36.96 | 42.04 |

Table 1: Performance of MT on MuST-C testset (BLEU↑). The input is the ground truth of the source transcription.

| Model | | En-De v2 | | En-De | | En-Fr | | En-Es | | Average Gap |
|---|---|---|---|---|---|---|---|---|---|---|
| | | common | he | common | he | common | he | common | he | |
| **Zero-Shot: MT is only trained on WMT corpus.** | | | | | | | | | | |
| **MultiSLT** | cascade | / | / | 17.30 | / | 27.15 | / | 21.29 | / | -13.79 |
| | zero-shot | / | / | 6.77 | / | 10.85 | / | 6.75 | / | |
| **Chimera** | zero-shot | / | / | 13.5 | / | 22.2 | / | 15.3 | / | / |
| **Ours** | cascade | 22.85 | 22.27 | 22.45 | 22.30 | 32.60 | 31.65 | 26.14 | 31.55 | +0.79 |
| | zero-shot* | 24.00 | 23.04 | 23.41 | 22.94 | 33.65 | 32.25 | 26.48 | 32.32 | |
| ***Pseudo* Zero-Shot: MT is trained on WMT and MuST-C parallel corpus.** | | | | | | | | | | |
| **Tight Integrated**[†] | cascade | / | / | 25.9 | 25.0 | / | / | 30.2 | 37.6 | -1.325 |
| | p. zero-shot | / | / | 25.1 | 24.4 | / | / | 28.7 | 35.2 | |
| **Ours** | cascade | 26.43 | 25.14 | 25.21 | 25.32 | 34.53 | 32.63 | 29.15 | 34.68 | +0.78 |
| | p. zero-shot* | 27.39 | 26.46 | 26.52 | 25.46 | 35.34 | 33.66 | 29.46 | 35.05 | |

Table 2: Zero-Shot ST on MuST-C (BLEU↑). [†]Tight Integrated extends our ASR data to 2300 hours, and it used 27M En-De and 48M En-Es MT data.

the ASR performance greatly affects the quality of CTC paths (Fan et al., 2020), our shrink method differs from previous approaches, where the adapter merges the representations from both before and after the CTC module to reduce error propagation. $\mathbf{h}_a$ can be regarded as the final audio representation which is ready to be fed into the semantic encoder. To alleviate the cross-modal mismatch, we optimize the WRD loss.

$$\mathbf{e}_a = \text{Enc}_t(\mathbf{h}_a) = \{\mathbf{e}_1^a, \ldots, \mathbf{e}_n^a\} \quad (2)$$
$$\mathbf{e}_t = \text{Enc}_t(\mathbf{h}_t) = \{\mathbf{e}_1^t, \ldots, \mathbf{e}_m^t\} \quad (3)$$
$$\mathcal{L}_{wrd} = D_{wrd}(\mathbf{e}_a, \mathbf{e}_t). \quad (4)$$

The detailed WRD loss is defined as follows.

$$D_{wrd}(\mathbf{e}_a, \mathbf{e}_t) = \langle \mathbf{C}, \mathbf{T}^* \rangle, \quad \mathbf{C}_{i,j} = 1 - \cos(\mathbf{e}_i^a, \mathbf{e}_j^t)$$

where $\langle \cdot, \cdot \rangle$ denotes the dot-product and $\cos(\cdot, \cdot)$ is the cosine similarity. $\mathbf{T}^*$ is the optimal transport (OT) plan from the following problem.

$$\mathbf{T}^* = \arg\min_{\mathbf{T} \geq 0} \langle \mathbf{C}, \mathbf{T} \rangle \quad \text{s.t.,} \mathbf{T}\mathbf{1}_m = \mathbf{p}, \mathbf{T}^\top \mathbf{1}_n = \mathbf{q}$$

where $\mathbf{p}, \mathbf{q}$ are the normalized $\mathbf{e}_a, \mathbf{e}_t$. Particularly, we represent them as two vectors $[p_1, \ldots, p_n]^\top$ and $[q_1, \ldots, q_m]^\top$.

$$p_i = \frac{\|\mathbf{e}_i^a\|_2}{\sum_{i=1}^{n} \|\mathbf{e}_i^a\|_2}, \qquad q_j = \frac{\|\mathbf{e}_j^t\|_2}{\sum_{j=1}^{m} \|\mathbf{e}_j^t\|_2}.$$

WRD emphasizes the semantic similarity between two sequences better than Euclidean distance. The common solution is to implement the Inexact Proximal point method for Optimal Transport (IPOT) algorithm (Xie et al., 2020) as shown in Appendix A.1. Because of CTC prediction errors, it is possible that $m \neq n$, but the loss $\mathcal{L}_{wrd}$ can circumvent the length discrepancy for alignment. The final loss of the cross-modal training is,

$$\mathcal{L}_{asr} = \lambda_{ctc}\mathcal{L}_{ctc} + \lambda_{wrd}\mathcal{L}_{wrd} \quad (5)$$

where $\lambda_{ctc}$ and $\lambda_{wrd}$ are hyper-parameters. To keep the semantic encoding intact, the semantic encoder including the embedding matrix is frozen, leading to a **zero-shot** translation system naturally from the ASR training.

### 2.3 End-to-End Translation Training

Once we have the triplets supervision dataset $\mathcal{D}_{Tri} = \{(\mathbf{z}, \mathbf{x}, \mathbf{y})\}$ such as speech-transcription-translation, it is possible to proceed for the fine-tuning phase as shown in Figure 1(e). Since the zero-shot training loss Eq. (5) is still valid in this phase, we can integrate it into the final end-to-end ST training objective $\mathcal{L} = \mathcal{L}_{st}(\mathbf{z}, \mathbf{y}) + \mathcal{L}_{asr}$.

## 3 Experiments

### 3.1 Datasets and Settings

**ST** We conduct our experiments on the three popular language pairs in MuST-C (Cattoni et al., 2021):

| Model | Num. Params | En-De v2 | | En-De | | En-Fr | | En-Es | |
|---|---|---|---|---|---|---|---|---|---|
| | | common | he | common | he | common | he | common | he |
| MTL[†] (Tang et al., 2021b) | 31M | / | / | 23.9 | / | 33.1 | / | 28.6 | / |
| FAT-ST (Zheng et al., 2021) | 58M | / | / | 25.5 | / | / | / | 30.8 | / |
| JT-S-MT[#] (Tang et al., 2021a) | 74M | / | / | 26.8 | / | 37.4 | / | 31.0 | / |
| Chimera (Han et al., 2021) | 165M | / | / | 27.1 | / | 35.6 | / | 30.6 | / |
| XSTNET (Ye et al., 2021) | 155M | / | / | 27.8 | / | 38.0 | / | 30.8 | / |
| STEMM (Fang et al., 2022) | 155M | / | / | **28.7** | / | 37.4 | / | 31.0 | / |
| **FT from zero-shot**[*] | 95M | **29.22** | 29.07 | 28.22 | 28.22 | 39.00 | 37.06 | 31.96 | 38.83 |
| **FT from p. zero-shot**[*] | 95M | 29.12 | **29.74** | 28.17 | 28.19 | **39.05** | **37.21** | **32.03** | **38.89** |

Table 3: Supervised ST on MuST-C (BLEU↑ with beam=5) with additional Librispeech and WMT data. [†] MTL uses the hidden dimension 256. [#] JT-S-MT only uses WMT data. [*] FT from our models marked with [*] in Table 2.

English-German (En–De) V1 and V2, English-French (En–Fr), and English-Spanish (En–Es).

**ASR** The 960h LibriSpeech English ASR dataset (Panayotov et al., 2015) is mainly used for pre-training the ASR in the zero-shot training stage.

**MT** For En-De and En-Fr, we collect the WMT 2014 data with about 4.5M and 36M parallel sentences respectively as in Vaswani et al. (2017). For En-Es, we collect the WMT 2013 data of size 28M.

**Model Details** The audio inputs are preprocessed as 80-channel log Mel filterbank coefficients as fairseq[1]. The cross-modal encoder contains two 1D convolutional subsampler layers (Synnaeve et al., 2019) and 12 transformer encoder layers with hidden dimension 512. The MT model is a standard transformer-based architecture (Vaswani et al., 2017). The individual vocabulary including 10K sub-word units is learned by Sentence-Piece (Kudo and Richardson, 2018). All hyper-parameters such as $\lambda$ are tuned on En-De V2 and directly applied to other datasets. Additional details can refer to the Appendix A.2.

## 3.2 Main Results

**Zero-shot ST** Recent works (Dinh, 2021; Escolano et al., 2021) indicate that when large amounts of ASR and MT data dominate the training, the cascaded ST is better than the direct end-to-end ST. In our proposed second phase, the desired ASR training can easily facilitate the building of a zero-shot ST model. The MT model is pre-trained with the WMT data alone, preventing the model from accessing the in-domain data of MuST-C. For the ASR training, we combine the Librispeech data and the speech-transcription pairs in MuST-C to give a comparable amount of ASR data as in the practical cascade system. Particularly, we set $\lambda_{ctc} = 1$ and

$\lambda_{wrd} = 10$.

In Table 1, we list the BLEU scores of our pre-trained MT models of the first training stage, in both zero-shot and pseudo zero-shot settings. Our main results of zero-shot ST are illustrated in Table 2. We compare our model with the pioneering zero-shot ST method **MultiSLT** (Escolano et al., 2021), achieving the zero-shot translation via ASR training with an adapter as well. We compare to another cross-modal alignment method **Chimera** (Han et al., 2021), which is initially designed for supervised ST training but also suitable for zero-shot ST. Clearly, our system can achieve a minimum gap between the cascade and the end-to-end setups in zero-shot scenario, and our end-to-end zero-shot ST on average performs +0.79 higher than that of the cascade system.

Following **Tight Integrated** (Dalmia et al., 2021; Bahar et al., 2021), we also conduct a *pseudo* zero-shot ST experiment. In this case, even though each training phase does not directly consume any speech-translation pairs, the overlapped MuST-C transcription data could be seen by both ASR and MT models. The gap between cascade and end-to-end remains unchanged (+0.79 → +0.78) for our model. It is an indication of the stability of our approach to bridging the modality gap.

**Supervised ST** In this experiment, we evaluate the performance of our third training phase. We compare our approach only in the unconstrained scenario with the recent end-to-end ST methods that used similar datasets, and the results are summarized in Table 3.

## 3.3 Analysis

For zero-shot ST setting, the average gap is 0.79, and the breakdown difference is visualized in the middle panel of Figure 2. The cascade system only has more sentences falling in BLEU interval [0, 10].

---

[1] https://github.com/facebookresearch/fairseq/tree/main/examples/speech_to_text

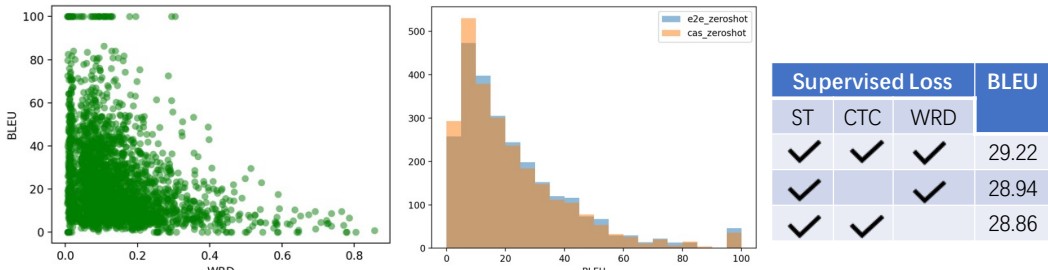

Figure 2: En-De v2. Left: BLEU *v.s.* WRD. Middle: Distribution of BLEU scores. Right: Ablation study of loss.

| Supervised Loss | | | BLEU |
|:---:|:---:|:---:|:---:|
| ST | CTC | WRD | |
| ✓ | ✓ | ✓ | 29.22 |
| ✓ | | ✓ | 28.94 |
| ✓ | ✓ | | 28.86 |

| Model Module | | | En-De v2 |
|:---:|:---:|:---:|:---:|
| Acoustic Encoder† | Adaptor | Semantic Enc/Dec* | common |
| trainable | trainable | trainable | 29.22 |
| trainable | trainable | frozen | 25.14 |
| frozen | trainable | trainable | 28.94 |

Table 4: Ablation study on the model parameters for supervised training phrase. All models are fine-tuned from the zero-shot ST model with BLEU 24.00 in Table 2. †Acoustic encoder includes the CTC layer.

We also plot the relation between BLEU and WRD for each sentence in the tst-CMMON set of En-De v2 (left panel of Figure 2). The overall trend indicates the BLEU score decreases with increasing word rotator's distance.

To achieve the zero-shot speech translation, the two losses in ASR training are both required. So the ablation study in the right panel of Figure 2 explored the effect of each loss in final end-to-end supervised training. All models are fine-tuned from the zero-shot ST model with BLEU 24.00 in Table 2. The CTC loss cannot be directly removed since the WRD depends on a reasonable CTC path. Therefore, we optimize the supervised loss without CTC loss by freezing the acoustic encoder and CTC layer.

In Table 4, we have another ablation study on whether the model parameters are trainable for the supervised training phase. The result becomes much worse if the semantic encoder/decoder is frozen. The main reason we hypothesize is that since the NMT teacher is frozen, the in-domain MT data is not used. So it's difficult for the NMT decoder to adapt to the supervised ST data, i.e., the decoder is not a good language model.

## 4 Conclusion

In this paper, we present a zero-shot architecture that takes better advantage of cascade models, bridging the gap between cascade and end-to-end translation models. By leveraging differentiable shrink adapter and WRD loss, our approach is a direct end-to-end ST model in the zero-shot setup that matches the cascade system without additional post-processing, e.g., rescoring via an additional language model. Our method can also achieve comparable results to recent supervised ST models.

## Limitations

The accuracy of IPOT depends on the number of loops. Since we unroll the IPOT similar to RNN training and apply automatic differentiation on the IPOT algorithm in the back-propagate stage, the iterative process will consume more computing resources than directly calculating the Jacobian matrix of the OT plan. Besides, though our model is able to work on different translation tasks such as image translation and speech translation, the hyper-parameters, especially the weights of WRD loss and CTC loss, would be varied on each task. The CTC loss and WRD loss are sometimes conflicting, which also requires us to set a pair of proper weights via deep hyper-para searching.

## Ethics Statement

After careful review, to the best of our knowledge, we have not violated the ACL Ethics Policy.

## Acknowledgements

We would like to thank all the anonymous reviewers for their insightful and helpful comments. This work was supported by Alibaba Group through Alibaba Research Intern Program.

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

# A Appendix

## A.1 WRD based IPOT

---

**Algorithm 1:** WRD based IPOT

---

**Input:** Maximum iterations $T = 50$,
encoded sequences
$\{\mathbf{e}_i^a\}_{i=1}^n, \{\mathbf{e}_j^t\}_{j=1}^m.$

1 Initialize $\mathbf{p}$ with $p_i = \frac{\|\mathbf{e}_i^a\|_2}{\sum_{i=1}^n \|\mathbf{e}_i^a\|}$

2 Initialize $\mathbf{q}$ similar to $\mathbf{p}$.

3 Initialize $\mathbf{C}$ as $\mathbf{C}_{i,j} = 1 - \cos(\mathbf{e}_i^a, \mathbf{e}_j^t)$.

4 $\mathbf{T} = \mathbf{1}_n \mathbf{1}_m^\top.$

5 $\boldsymbol{\sigma} = \frac{1}{m} \mathbf{1}_m, \mathbf{G}_{i,j} = e^{-\mathbf{C}_{i,j}}.$

6 **for** $t = 1, 2, \dots, T$ **do**

7 $\quad \mathbf{Q} = \mathbf{G} \odot \mathbf{T}$

8 $\quad \boldsymbol{\delta} = \frac{\mathbf{p}}{\mathbf{Q}\boldsymbol{\sigma}}, \boldsymbol{\sigma} = \frac{\mathbf{q}}{\mathbf{Q}^\top \boldsymbol{\delta}}$

9 $\quad \mathbf{T} = \text{diag}(\boldsymbol{\delta})\mathbf{Q}\text{diag}(\boldsymbol{\sigma})$

10 **return** $\langle \mathbf{C}, \mathbf{T} \rangle$

---

IPOT replaces the Bregman divergence $D_h(\mathbf{x}, \mathbf{y}) = \sum_{i=1}^n x_i \log \frac{x_i}{y_i} - \sum_{i=1}^n x_i + \sum_{i=1}^n y_i$ with the proximal point iteration, *i.e.*, substitutes the following iterative update for the original optimization problem.

$$\mathbf{T}^{(t+1)} = \arg \min_{\mathbf{T} \geq 0} \langle \mathbf{C}, \mathbf{T} \rangle + \beta^{(t)} D_h(\mathbf{T}, \mathbf{T}^{(t)}) \quad (6)$$

Algorithm 1 shows the detailed implementation of IPOT, where $\text{diag}(\boldsymbol{\delta})$ represents the diagonal matrix with $\delta_i$ as its $i$-th diagonal element, and $\odot$ and $\frac{(\cdot)}{(\cdot)}$ denote the element-wise matrix multiplication and division respectively. The algorithm outlines the forward-propagation steps only. Since each iteration of the algorithm only involves differentiable operators, we can utilize the automatic

differentiation packages (*e.g.*, `PyTorch`) to back-propagate the gradients like an unrolled RNN. The corresponding implementation can refer to the submitted software.

## A.2 Additional Training Details

As model design in Figure 1, the embedding layer in the adapter shares the weights with the semantic source embedding. The beam size is 5 during inference, and we use SacreBLEU in fairseq as evaluation metrics.

For the third phase, supervised ST training, we have multiple tasks in the final objective. For the ST task $\mathcal{L}_{st}$, some previous works may leverage the MT model and the Librispeech transcription to construct pseudo translation sentences. However, we only use the audio and translation pairs from MuST-C. For the ASR task $\mathcal{L}_{ctc}$, we only use the audios and transcriptions from MuST-C. For the MT task $\mathcal{L}_{mt}$, we optimize it on both the MuST-C parallel corpus and WMT data, making the decoder a better language model. En-De WMT only has 4.5M sentence pairs and the entire training is still manageable. However, for En-Fr/Es, optimizing the large end-to-end ST model with a huge amount of trainable parameters will be cumbersome because the size of WMT data overwhelmingly slows down the training. Therefore, we randomly sample a 10M corpus from the original WMT En-Fr/Es data to train the final supervised loss.

## A.3 Additional Experimental Results of ST

In Figure 3, we plot the WER scores of the second training stage, in both zero-shot and pseudo zero-shot settings. The ASR of the cascade system (*i.e.*, trained with CTC loss only and without semantic encoder) has a clearly higher WER than our proposed ASR training with additional WRD loss. However, the in-domain MuST-C data do not appear to make a significant difference as indicated by the orange and the green bars in Figure 3.

## A.4 Generalization and Visualization

We also conduct an experiment on zero-shot Image Translation using only OCR data and NMT data to further test the effectiveness of our framework. It is also convenient to visualize with image data. The NMT model (*i.e.* the semantic encoder and decoder) is pre-trained on the WMT 2018 Zh-En data (20M) parallel sentences in the news domain. We crop 2M text-line images from Chinese OCR

data [2]. The test set has 2,000 images with Chinese transcriptions and English translations. The BLEU on the test set for the pre-trained NMT is 15.87, which is not high due to the domain shift.

In particular, we set different weights $\lambda_{wrd} = 0, 1, 5, 10, 20, 50$ to investigate the effectiveness of the WRD loss, where the model with $\lambda_{wrd} = 0$ reduces to a cascade model. The results of the zero-shot Image Translation are shown in Figure 4. It illustrates the intuition of how to tune the importance of WRD loss. In Figure 5, we visualize the transport plan $\mathbf{T}^*$ and cost matrix $\mathbf{C}$ of some examples.

---

[2] https://github.com/YCG09/chinese_ocr and https://taobao.com

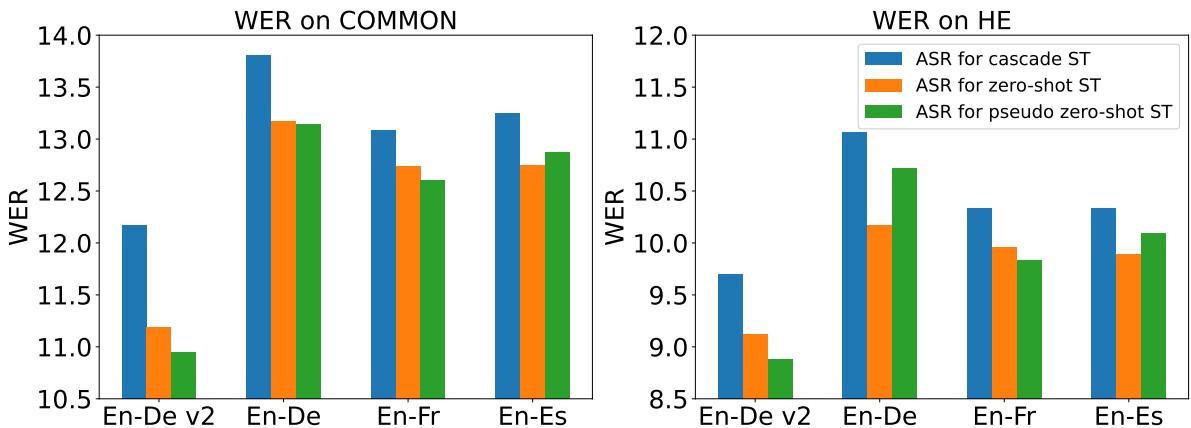

Figure 3: The performance of the ASR as zero-shot ST systems.

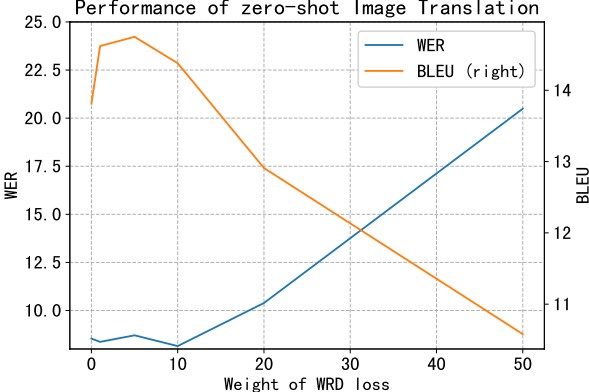

Figure 4: The performance of OCR and zero-shot image translation over different weights $\lambda_{wrd}$.

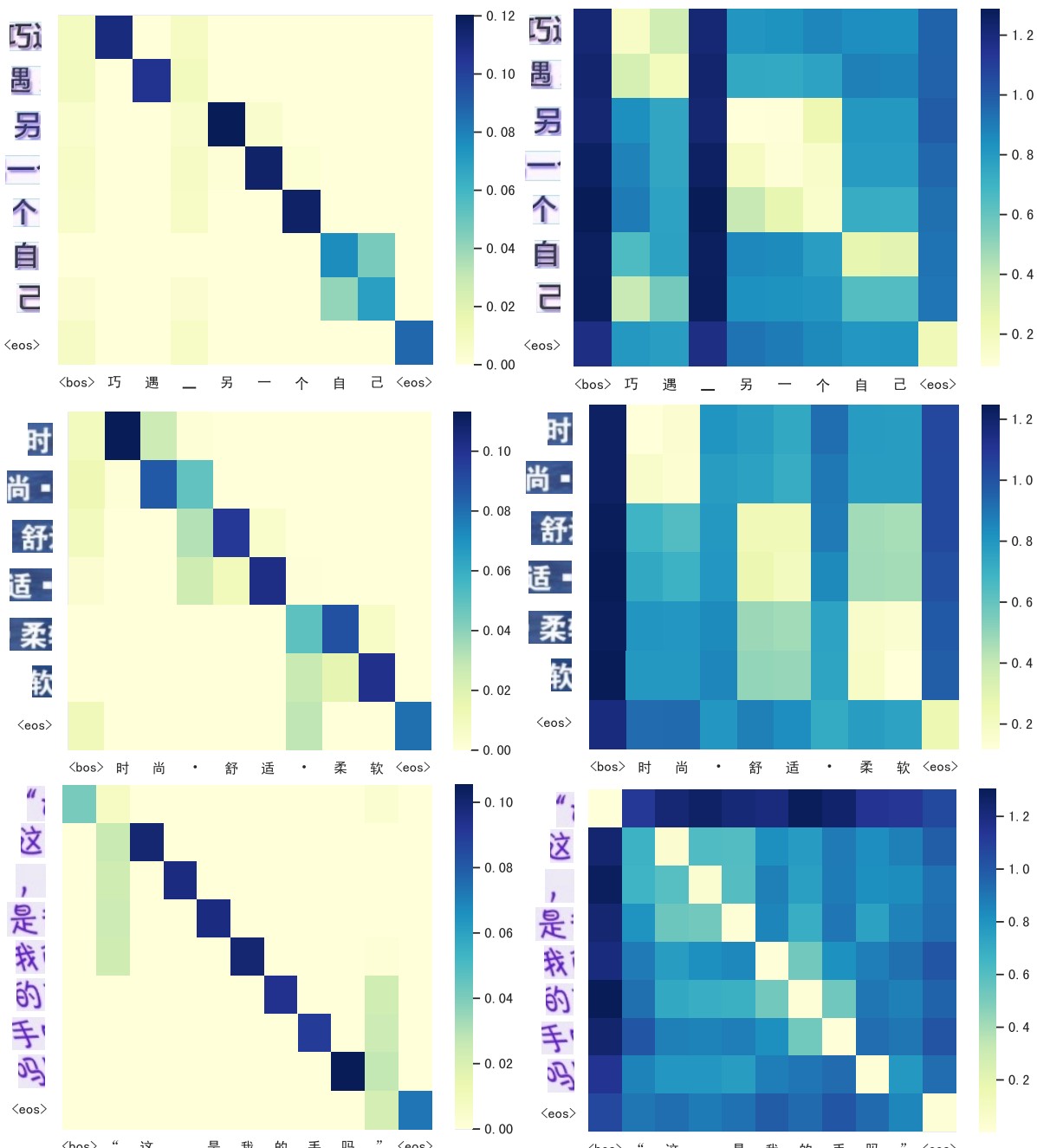

Figure 5: **Left Column**: visualization of transport plan $\mathbf{T}^*$. **Right Column**: visualization of cost matrix $\mathbf{C}$. The WRD with OT solver can align cross-modal features with different lengths. $y$-axis represents the shrunk features, and $x$-axis represents the transcription features. Since the actual shrunk tokens only represent a 4-pixel wide part of an image, we cut images along blank tokens as a schematic representation. The real shrunk tokens are usually in the middle of the images on $y$-axis. For the cost matrix, the smaller elements are mainly distributed on the diagonal block regions. It could be the incorrect shrinking segments sometimes aligning with more than one character. For the transport plan matrix, the larger elements are mainly distributed on the diagonal. In this way, their products will remain small.