# OpenReview forum: "Towards Zero-shot Learning for End-to-end Cross-modal Translation Models"
_EMNLP/2023/Conference — EMNLP 2023 Findings_

### Official Review · Reviewer_ZEUx · 2023-08-01

**Typos Grammar Style And Presentation Improvements:** Line 248 has missing table reference.
**Soundness:** 3

**Excitement:**

4: Strong: This paper deepens the understanding of some phenomenon or lowers the barriers to an existing research direction.

**Paper Topic And Main Contributions:**

This paper proposes a novel method for end to end speech translation using only machine translation (MT) and automatic speech recognition (ASR) data in zero shot setting as well as same in fine-tune setting with speech translation (ST) data. For zero shot setting ASR and MT models are trained by using Word Rotator Distance (WRD) to align transcribed representation of speech with translation model in conjuction with usual CTC loss on CTC based speech encoder. For fine tune setting translation model can be trained with source transcripts and target transcripts of ASR data.

**Questions For The Authors:**

Question A: Certain design choices are not clear to me. AFAIK best path taken during CTC training can be from all possible CTC alignments instead of being just argmax of choices present at current stage? Why authors particularly chose argmax?
Question B: I felt comparison with cascade based models in which end translation decoder trained end to end would have been more fair.
Question C: I felt comparison of latency with earlier cascade based models could have been good to bring out novelty better.

**Reasons To Accept:**

1. I really like how end to end model using models from two different domains ASR and MT aligned by word rotator distance.
2. Problem formulation is well defined and afaik results seems to perform better or as good as current CTC cascade based models without any added overhead of existing cascade models in terms of latency.



**Reasons To Reject:**

1. Paper is hard to follow sometimes. I felt paper has missed some important reference. Like CTC is mentioned multiple times like in line 019, 069, 083 but no reference is provided making it difficult for readers from different domain to follow. Second ST dataset is not properly described in section 3.1 leading to confusion what common and he means in Table 1,2.
2. For the reported results in table 1 and table 2, how results are obtained for prior work is not clear to me. Is it ensured that all experiments are performed in same settings as in prior works?

**Reproducibility:**

4: Could mostly reproduce the results, but there may be some variation because of sample variance or minor variations in their interpretation of the protocol or method.

**Reviewer Confidence:**

3: Pretty sure, but there's a chance I missed something. Although I have a good feel for this area in general, I did not carefully check the paper's details, e.g., the math, experimental design, or novelty.

---

> ### Author Rebuttal · Authors · 2023-08-29
>
> Thanks for the positive comments. We will include the suggested references and describe the details of dataset in the revised version. For the results in Table 1 and 2, we tried our best to keep the experimental setting the same. In the caption of each Table, we've marked the difference against our setting. However, not all papers release their code and checkpoint, so the marks in the caption are mainly based on the description in the original papers.
>
> **Questions**
> >Question A: Certain design choices are not clear to me. AFAIK best path taken during CTC training can be from all possible CTC alignments instead of being just argmax of choices present at current stage? Why authors particularly chose argmax?
>
> Thanks for the great suggestion. It's definitely correct and better to use the best CTC path from all possible alignments. In our work, we use $\arg\max$ for efficiency. Because $\arg\max$ is more friendly for parallelization, leading to faster training speed. In contrast, even dynamic programming is used in searching all possible paths is not efficient for training purpose. However, we appreciate this suggestion and will explore its possibility for inference.
>
> >Question B: I felt comparison with cascade based models in which end translation decoder trained end to end would have been more fair.
>
> We apologized that we are not sure what exactly this question means. The cascade system includes two independent models: ASR and NMT. The ASR is trained on audio-transcription data, and the NMT is an encoder-decoder architecture trained on parallel (source-target) text corpus. In the zero-shot setting, the source text in MT data and the transcription in ASR data have no overlap. Due to the unavailability of audio-target data, the translation decoder cannot be trained end-to-end.
>
> Our explanation is based on our understanding, and we are open to provide more response during the discussion window.
>
> >Question C: I felt comparison of latency with earlier cascade based models could have been good to bring out novelty better.
>
> Thanks for this suggestion. As we mentioned earlier, it is difficult for us to access earlier cascade based models for either code repository or checkpoint. Similarly, for the end-to-end models, it is also difficult to access their resources, so we list the number of parameters.
>
> We submitted our code as the supplementary material and will open-source our code upon acceptance. We would like to add the latency of our own model in the revised version.

---

### Official Review · Reviewer_kgo5 · 2023-08-04

**Typos Grammar Style And Presentation Improvements:** In 3.3 table ref is shown as ???
**Soundness:** 2

**Excitement:**

3: Ambivalent: It has merits (e.g., it reports state-of-the-art results, the idea is nice), but there are key weaknesses (e.g., it describes incremental work), and it can significantly benefit from another round of revision. However, I won't object to accepting it if my co-reviewers champion it.

**Paper Topic And Main Contributions:**

The paper proposed an E2E speech translation model designed for a zero-shot setting, considering the scarcity of parallel data during training. Specifically, the speech translation model is constructed by integrating the word rotator's distance loss to connect two pretrained uni-modality parts: ASR (Automatic Speech Recognition) and the Machine Translation (MT) modules.


**Questions For The Authors:**

A. As mentioned in the weaknesses of the paper, it is understandable that the selected submodules in the cascade models may not be the optimal ones. However, it is evident that the chosen modules are significantly inferior compared to the state-of-the-art (SOTA) counterparts. This raises concerns about the effectiveness of the proposed E2E (End-to-End) model.
If we were to employ the SOTA submodules from ASR and MT, along with a much larger amount of data within each modality, the question arises: do the advantages of the proposed method still hold true?
In addition, the reference and more detailed description of each submodule is missing, which makes it hard in terms of reproducibility.

B. In each table, what is the common and he? please clarify.

C. While the authors present the ablation study in the table shown in Figure 2, the observed differences between the results may be small. The question arises whether the authors conducted any significance tests to determine the statistical significance of these differences.

**Reasons To Accept:**

This paper proposes the utilization of the Word Rotator's Distance loss and a shrink adapter to address the multi-modal discrepancy, which is the major novelty of this paper. Interesting results are shown to convince the effectiveness of the method.

**Reasons To Reject:**

The paper is lacking crucial details, and proving the generalization of the proposed method might be difficult due to the selection of different submodules (as indicated in the questions below).

For example, the ASR system, based on the limited description in the paper, utilizes a CTC loss with some transformer encoder layers, which is considerably weaker compared to the state-of-the-art (SOTA) ASR modules. For instance, using Conformer layers in the encoder has demonstrated significantly better performance on the spectrum input. Furthermore, relying solely on CTC without a language model can result in poor ASR output performance. Similar concerns apply to the MT (Machine Translation) sub-system.

Considering the comparable performance or slightly better performance on the proposed e2e settings and the selection of submodules, it could be hard to demonstrate the effectiveness of its proposed approach.
In addition, based on the results in the table (advantages compared to other existing work), experiments are not convincing enough.

**Reproducibility:**

2: Would be hard pressed to reproduce the results. The contribution depends on data that are simply not available outside the author's institution or consortium; not enough details are provided.

**Reviewer Confidence:**

4: Quite sure. I tried to check the important points carefully. It's unlikely, though conceivable, that I missed something that should affect my ratings.

---

> ### Author Rebuttal · Authors · 2023-08-29
>
> Thanks for the constructive comments. We will fix the typos in the revised version. The concerns will be addressed first, then the question will be responded.
>
> >Concern 1: SOTA ASR and MT are not used in cascade system.
>
> Thanks for raising this issue. In the research line of zero-shot end-to-end ST, the main purpose is not to compare with a cascade system with both SOTA ASR and NMT models. As previous works (e.g., MultiSLT [1], DCMA [2]), similar architecture and similar model size between cascade and end-to-end systems are usually adopted. In this way, two systems can be fairly compared, e.g., the paper "Cascade versus Direct"[3] applied systematic comparison between two systems in similar transformer architecture without language model rescoring. In summary, training two systems to approach SOTA but with different architectures and different datasets did not fall into this research area.
>
> [1] MultiSLT: Enabling Zero-Shot Multilingual Spoken Language Translation with Language-Specific Encoders and Decoders, (ARSU 2021)
>
> [2] DCMA: Discrete Cross-Modal Alignment Enables Zero-Shot Speech Translation, (EMNLP 2022)
>
> [3] Cascade versus Direct: Cascade versus Direct Speech Translation: Do the Differences Still Make a Difference? (ACL 2021)
>
> >Concern 2: Considering the comparable performance or slightly better performance on the proposed e2e settings, it could be hard to demonstrate the effectiveness.
>
> Thanks for proposing this concern. We will like to address our title or main contribution is exploring the zero-shot end-to-end ST system, and the main result is shown in Table 1. We extracted the the last column from Table 1 in our paper.
> || Average GAP|
> |:----------------|:----:|
> |previous SOTA|-13.79 / -1.325|
> |ours|+0.79 / +0.78|
>
> **GAP**: BLEU of zero-shot end-to-end system $-$ BLEU of zero-shot cascade system.
> We can clearly see our approach achieve significant improvement to close the gap between zero-shot cascade and end-to-end systems.
>
> **Questions**
> >A. As mentioned in the weaknesses of the paper, it is understandable that the selected submodules in the cascade models may not be the optimal ones. However, it is evident that the chosen modules are significantly inferior compared to the state-of-the-art (SOTA) counterparts. This raises concerns about the effectiveness of the proposed E2E (End-to-End) model. If we were to employ the SOTA submodules from ASR and MT, along with a much larger amount of data within each modality, the question arises: do the advantages of the proposed method still hold true? In addition, the reference and more detailed description of each submodule is missing, which makes it hard in terms of reproducibility.
>
> As we explained in the response for concerns, utilizing the SOTA ASR and MT is not the main objective of this area. Our current adapter is designed based on two commonly used ASR and MT architectures,  a CTC-based model and a transformer encoder-decoder model. If similar architecture is used, our proposed adapter should be still valid, e.g., changing the transformer layer to conformer layer. If we use a SOTA model with very different architecture, we may have to re-design the adapter accordingly.
>
> For the reproducibility, we submitted the code and we will open-source our code upon acceptance. The detailed implementation is in the code folder "examples/dcm/". We appreciate your suggestions and will introduce more details in the revised version.
>
> >B. In each table, what is the common and he? please clarify.
>
> In the widely used ST dataset MuST-C, there are two testsets for each language pair. The dataset names of the two testsets are "common" and "he", where "common" means common talks and "he" means an additional testset.
>
> >C. While the authors present the ablation study in the table shown in Figure 2, the observed differences between the results may be small.
> The question arises whether the authors conducted any significance tests to determine the statistical significance of these differences.
>
> We did not conduct any significance tests on the ablation study in Figure 2, because our work did not focus on the supervised learning. Our main contribution is the zero-shot end-to-end learning, while supervised continue training is a by-product. Note that supervised continue learning is different from direct supervised learning (other methods shown in Table 2). In addition, the last column in Table 1 shows the gap or the improvement of our approach is significantly better than previous works in relevant research area.

---

### Official Review · Reviewer_aFGi · 2023-08-09

**Soundness:** 3

**Excitement:**

4: Strong: This paper deepens the understanding of some phenomenon or lowers the barriers to an existing research direction.

**Paper Topic And Main Contributions:**

This paper is about End-to-end Cross-modal Translation for Zero-shot.

Main contributions:They propose an end-to-end zero-shot speech translation model to adress the mismatches between different modalities and scarcity of parallel data. Adopting the WRD loss together with the shrink mechanism to measure two feature sequences in different lengths, enabling the adapter pre-training.


**Reasons To Accept:**

This paper proposes a new approach to address the mismatches between different modalities and scarcity of parallel data. Moreover, the experiments were conducted thoroughly, and the model diagrams were beautifully illustrated, making it a good short paper.

**Reasons To Reject:**

Table 2 presents a limited number of new end-to-end ST models for comparison.

**Reproducibility:**

4: Could mostly reproduce the results, but there may be some variation because of sample variance or minor variations in their interpretation of the protocol or method.

**Reviewer Confidence:**

3: Pretty sure, but there's a chance I missed something. Although I have a good feel for this area in general, I did not carefully check the paper's details, e.g., the math, experimental design, or novelty.

---

> ### Author Rebuttal · Authors · 2023-08-29
>
> Thanks for the insightful comment. We will address the main conern "Table 2 presents a limited number of new end-to-end ST models for comparison."
>
> As the main research area and contribution of our work is zero-shot end-to-end ST, it means the transcriptions in ASR dataset and the source sentences in MT dataset have no overlap (as shown in Figure (b)). However, most end-to-end ST models are not trained in the zero-shot setup. To our best knowledge, we have already included the relevant works in our paper.
>
> Supervised continue training is a by-product of our work, but most of them are not comparable with ours. Because they are directly trained in supervised or semi-supervised learning rather than supervised continue training from a zero-shot model.
>
> In Table 2, we have listed some recent supervised ST models with similar model size. We still appreciate your suggestions and will include more supervised ST in the revised version.

---

### Official Review · Reviewer_B6yQ · 2023-08-10

**Soundness:** 3

**Excitement:**

3: Ambivalent: It has merits (e.g., it reports state-of-the-art results, the idea is nice), but there are key weaknesses (e.g., it describes incremental work), and it can significantly benefit from another round of revision. However, I won't object to accepting it if my co-reviewers champion it.

**Paper Topic And Main Contributions:**

The authors propose a zero-shot learning method for end-to-end speech translation models. Unlike the cascade models that feed a token (word) sequence output from a speech recognition system to a text-based translation system, the proposed method connects the two systems using a variable-length embedding representation. We can expect robustness to speech recognition errors by using the embedding representation. Their method can train the embedding representation using independent speech recognition and machine translation training data, realizing zero-shot learning without requiring speech-transcript-translation triplet supervision data. It is also possible to fine-tune the system if triplet supervision data is available. The experiments show that the proposed method can match or be slightly better than the CTC-based cascade model. The results of end-to-end training also match the recent supervised speech translation baselines.

**Questions For The Authors:**

In Table 1:
Does "Ours cascade" mean the baseline system without the shrink adapter that uses a word sequence (text) to connect a speech recognizer and a translation system?

In Equation (2), e_a = {e_1^a, ..., t_n^a):
What is the difference between "e" and "t" in e_1^a and t_n^a?

In Abstract, "without introducing any trainable parameters":
Are the parameters in the Shrink Adapter not counted here?




**Reasons To Accept:**

The paper proposes a new method to connect a speech recognizer and a text translation system using an embedded representation that can work without requiring end-to-end data, which is often limited.

**Reasons To Reject:**

The text needs major revision for readability. It isn't easy to understand the details. They do not explain many of the variables and the Table entries.

**Reproducibility:**

3: Could reproduce the results with some difficulty. The settings of parameters are underspecified or subjectively determined; the training/evaluation data are not widely available.

**Reviewer Confidence:**

3: Pretty sure, but there's a chance I missed something. Although I have a good feel for this area in general, I did not carefully check the paper's details, e.g., the math, experimental design, or novelty.

**Typos Grammar Style And Presentation Improvements:**

Introduction "with OT solver:":
 The authors should describe full spelling when they first introduce an abbreviation.

Section 3.1:
The cross-model encoder: cross-modal (?)

Section 3.3:
Table ??

---

> ### Author Rebuttal · Authors · 2023-08-29
>
> Thanks for the insightful comments. In general, we will fix all typos and make the statement more clear in the revised version.
>
> >Q1: In Table 1: Does "Ours cascade" mean the baseline system without the shrink adapter that uses a word sequence (text) to connect a speech recognizer and a translation system?
>
> Yes. In the cascade system, the ASR (speech recognizer) and the NMT (translation system) have the same model architecture as our proposed end-to-end approach. Only the shrink adapter is removed. For fair comparison in the zero-shot setups, most previous works (e.g., MultiSLT [1], DCMA [2], Cascade versus Direct[3]) usually adopt similar architecture and similar model size between cascade and end-to-end systems.
>
> [1] MultiSLT: Enabling Zero-Shot Multilingual Spoken Language Translation with Language-Specific Encoders and Decoders, (ARSU 2021)
>
> [2] DCMA: Discrete Cross-Modal Alignment Enables Zero-Shot Speech Translation, (EMNLP 2022)
>
> [3] Cascade versus Direct: Cascade versus Direct Speech Translation: Do the Differences Still Make a Difference? (ACL 2021)
>
> >Q2: In Equation (2), $e_a = (e_1^a, ... ,t_n^a)$ What is the difference between $e$ and $t$ in $e_1^a$ and $t_n^a$?
>
> Sorry, this is a typo. It should be $e_a = (e_1^a, ..., e_n^a)$. We will fix this typo in the revised version.
>
> >Q3: In Abstract, "without introducing any trainable parameters": Are the parameters in the Shrink Adapter not counted here?
>
> Sorry for the unclear description. We originally mean that unlike other approaches that introduce additional trainable parameters during the end-to-end training stage, our adapter has already been pre-trained. Therefore, no additional parameters will be introduced in the fine-tuning stage. We will clarify this statement in the revised version.

---

### Meta-Review · Area_Chair_RMWJ · 2023-09-18

**Recommendation:** 4

**Metareview:**

This paper proposes a zero-shot learning approach that bridges the gap between cascade and end-to-end speech translation models using a differentiable shrink adapter and WRD loss.

The reviewers' evaluations are mostly consistent, showing moderate soundness and relatively high excitement scores.
The reviewers are concerned about the readability to understand the details, limited comparisons, and the lack of SOTA ASR system, e.g., Conformer-CTC with an external language model.

The authors answered the reviewers' questions regarding technical details, and the answers sound reasonable to me.
The authors also explained that the focus of the paper is to bridge the gap between the cascade and end-to-end systems and not to achieve SOTA performance.
This is understandable, but I think that evaluating using Conformer-CTC with a language model would have yielded more reliable results. However, the topic the paper addresses is interesting. The reviewers rated high excitement scores. So, it will be of interest to a large audience. If accepted, I hope that the authors will try to improve the readability as pointed out by the reviewers.

---

### Decision · Program_Chairs · 2023-10-07

**Decision:**

Accept-Findings

**Comment:**

This paper proposes a zero-shot learning approach that bridges the gap between cascade and end-to-end speech translation models using a differentiable shrink adapter and WRD loss.

The reviewers' evaluations are mostly consistent, showing moderate soundness and relatively high excitement scores.
The reviewers are concerned about the readability to understand the details, limited comparisons, and the lack of SOTA ASR system, e.g., Conformer-CTC with an external language model.

The authors answered the reviewers' questions regarding technical details, and the answers sound reasonable to me.
The authors also explained that the focus of the paper is to bridge the gap between the cascade and end-to-end systems and not to achieve SOTA performance.
This is understandable, but I think that evaluating using Conformer-CTC with a language model would have yielded more reliable results. However, the topic the paper addresses is interesting. The reviewers rated high excitement scores. So, it will be of interest to a large audience. If accepted, I hope that the authors will try to improve the readability as pointed out by the reviewers.